# Development of a Bioactive Flowable Resin Composite Containing a Zinc-Doped Phosphate-Based Glass

**DOI:** 10.3390/nano10112311

**Published:** 2020-11-22

**Authors:** Myung-Jin Lee, Young-Bin Seo, Ji-Young Seo, Jeong-Hyun Ryu, Hyo-Ju Ahn, Kwang-Mahn Kim, Jae-Sung Kwon, Sung-Hwan Choi

**Affiliations:** 1Department of Dental Hygiene, Division of Health Science, Baekseok University, Cheonan 31065, Korea; dh.mjlee@bu.ac.kr; 2Department and Research Institute of Dental Biomaterials and Bioengineering, Yonsei University College of Dentistry, Seoul 03722, Korea; youngbin@yuhs.ac (Y.-B.S.); sweetden623@gmail.com (J.-H.R.); hjan505@yuhs.ac (H.-J.A.); kmkim@yuhs.ac (K.-M.K.); 3BK21 FOUR Project, Yonsei University College of Dentistry, Seoul 03722, Korea; 4Department of Orthodontics, Institute of Craniofacial Deformity, Yonsei University College of Dentistry, Seoul 03722, Korea; jyseo13@yuhs.ac

**Keywords:** dental restoration, flowable resin composite, antibacterial, plaque prevention, biofilm, zinc, bioglass, bioactive material, tooth remineralization

## Abstract

Flowable resins used for dental restoration are subject to biofilm formation. Zinc has antibacterial properties. Thus, we prepared a zinc-doped phosphate-based glass (Zn-PBG) to dope a flowable resin and evaluated the antibacterial activity of the composite against *Streptococcus mutans* (*S. mutans*) to extrapolate the preventative effect toward secondary caries. The composites were prepared having 0 (control), 1.9, 3.8, and 5.4 wt.% Zn-PBG. The flexural strength, elastic modulus, microhardness, depth of cure, ion release, inhibition zone size, and number of colony-forming units were evaluated and analyzed using ANOVA. The flexural strength of the control was significantly higher than those of Zn-PBG samples (*p* < 0.05). However, all samples meet the International Standard, ISO 4049. The microhardness was not significantly different for the control group and 1.9 and 3.8 wt.% groups, but the 5.4 wt.% Zn-PBG group had a significantly lower microhardness (*p* < 0.05). Further, the composite resins increasingly released P, Ca, Na, and Zn ions with an increase in Zn-PBG content (*p* < 0.05). The colony-forming unit count revealed a significant reduction in *S. mutans* viability (*p* < 0.05) with increase in Zn-PBG content. Therefore, the addition of Zn-PBG to flowable composite resins enhances antibacterial activity and could aid the prevention of secondary caries.

## 1. Introduction

Flowable resin composites have been clinically proven to be an effective dental treatment over the last 60 years are a popular restorative material in dentistry because of the good flowability and color, which is similar to that of natural tooth enamel [1,2]. Flowable resin composites have a diverse variety of applications, including preventive resin restoration, pit and fissure sealing, restoration repair, and cavity lining [3]. Although flowable composite resins have shown long-term clinical success, they tend to accumulate more bacteria and dental plaque than enamel and other restorative materials [4]. Unfortunately, the accumulation of plaque on the restorative material increases the possibility of the development of secondary caries [5].

Secondary caries can lead to the weakening of the enamel around the restoration site, marginal breakdown, and loss of the restored surface [6,7]. Therefore, the development of restorative materials with antibacterial activity is necessary to reduce the risk of secondary caries. Antimicrobial agents have limited application because ion release and uptake diminish with time. Thus, they typically only show short-term effects [8]. Consequently, the development of an antimicrobial material having long-term, continuous action without antimicrobial resistance or adverse effects on humans and the environment is essential.

Zinc oxide (ZnO) has antimicrobial properties and low toxicity [9,10,11]. Therefore, zinc oxide has been extensively studied in the dental field for its ability to inhibit the growth of bacteria. In previous studies, zinc has been incorporated into dental materials using ZnO particles as fillers [10,12], and zinc has been added to toothpastes and mouth rinses to inhibit dental plaque and reduce halitosis [10]. However, achieving the continuous release of Zn ions is challenging, and this limits long-term antibacterial activity.

Bioglasses are glass-like materials with a composition containing more calcium and phosphate than silica glasses. Thus, their composition is more similar to bone and tooth. This makes them potential materials for use in dental repair. The first bioactive glass, 45S5 bioglass, was discovered by Larry Hench [13]. However, an alternate 45S5 bioglass for biomedical applications is phosphate-based glass (PBG) [14]. PBG is based on P_2_O_5_ as the glass network former but also contains CaO and Na_2_O. PBG has a long-term effect because of its slow degradation over 1 to 2 years [15]. PBG is suitable for use in biomaterial and tissue engineering because the chemical composition of glass is close to that of natural bone and teeth [16,17], and the calcium and phosphate contents of PBG bioactive glass play important roles in the remineralization process [18]. In particular, phosphate greatly contributes to hydroxyapatite (HA) formation and increases biocompatibility [19,20].

Previous studies have concentrated on the effect of adding zinc ions or bioactive glass to dental materials such as flowable composite resins. However, few studies have investigated the antimicrobial effect of the incorporation of Zn-PBG into flowable composite resins.

In this study, the antimicrobial properties of flowable composite resins with added Zn-PBG were investigated the suitability of the flowable composite containing Zn-PBG for use in dentistry. The objective of this study was to investigate the mechanical and ion releasing properties, as well as the antibacterial activities of bioactive flowable resin composites containing Zn-PBG. The null hypotheses of this study are that (1) a flowable resin composite containing Zn-PBG would not result in significant differences in the mechanical properties compared to the control, (2) a flowable resin composite containing Zn-PBG would not result in significant differences in P, Ca, and Zn ion release compared to the control, and (3) the flowable resin composite containing Zn-PBG would not result in significant differences in antibacterial properties compared to the control.

## 2. Materials and Methods

### 2.1. Glass Preparation

To obtain glass frit, batch P_2_O_5_ (42 mol.%), CaO (25.2 mol.%), Na_2_O (16.8 mol.%), and ZnO (16 mol.%) powders were well-mixed in a tubular shaker-mixer for 60 min, and the mixed batches was melted in an alumina crucible at 1100 °C for 1 h. Subsequently, the melted glass was quenched at room temperature to obtain a glass cullet. This glass cullet was coarsely ground in an alumina mortar and then pulverized under dry conditions using a planetary mono mill (Pulverisette-7; Fritsch, Idar-Oberstein, Germany). The particle size and morphology of the Zn-PBG were determined using histograms of the Zn-PBG particle size distribution obtained using a particle size analyzer (Mastersizer 2000, Malvern Instruments, UK) and a field-emission scanning electron microscope (FE-SEM, JSM-7800 F, JEOL, Akishima, Tokyo, Japan). Furthermore, the amorphous structure of Zn-PBG was confirmed by X-ray diffraction (XRD) analysis (Rigaku, Tokyo, Japan) between 2θ of 20° and 80°.

### 2.2. Incorporation of Zn-PBG Into the Flowable Composite Resin

A commercially available flowable composite resin (G-aenial Universal Flo; GC, Tokyo, Japan) was used in this study. Zn-PBG powder was mixed into the flowable composite resin at various weight percentages (1.9, 3.8, and 5.4 wt.%), and a flowable composite resin without Zn-PBG was used as a control. The compositions of the control and experimental materials are summarized in Table 1. Glass powder with a composition of P_2_O_5_ (42 mol.%), CaO (25.2 mol.%), Na_2_O (16.8 mol.%), and ZnO (16 mol.%) was prepared, and it was named Zn-PBG. Zn-PBG powder was mixed into the flowable composite resin at various weight percentages (1.9, 3.8, and 5.4 wt.%), and a flowable composite resin without Zn-PBG was used as a control (0 wt.%). All samples were prepared before polymerization. The Zn-PBG powder was mixed into the flowable composite resin by hand mixing. After mixing, the samples were polymerized using a light-emitting diode (LED) light-curing unit (Elipar S10; 3M ESPE Co., Seefeld, Germany).

### 2.3. Mechanical Properties

#### 2.3.1. Flexural Strength and Elastic Modulus

The flexural strength and elastic modulus of the flowable composite resin were tested according to the International Standard, ISO 4049 (2019). Five rectangular specimens with 25-mm length, 2-mm width, and 2-mm height were fabricated for each composite resin group. Each group of composite resin was polymerized with five slightly overlapping irradiation regions (20 s each) from both sides. Before flexural strength and elastic modulus testing, all samples were stored at 37 °C in distilled water for 24 h. The specimens were loaded to fracture by three-point bending using a universal testing machine (Model 5942, Instron, Norwood, MA, USA) at a span length of 20 mm and a crosshead speed of 1 mm/min. The flexural strength, *σ*, and elastic modulus, *E*, were calculated according to Equations (1) and (2), respectively.
(1)$$\sigma =\frac{3Fl}{2bh^{2}}$$
(2)$$E=\frac{Pl^{3}}{4bh^{3}d}$$
where *F* is the maximum load exerted (N), *l* is the distance between the supports (mm), *b* is the width, *h* is the height of the specimen (mm), *P* is the load at a point in the straight-line portion of the load/displacement curve (N), and *d* is the deflection at load *P* (mm).

#### 2.3.2. Microhardness

Five disc specimens having diameters and heights of 10 and 2 mm, respectively, were prepared for each composite resin group. The Vickers microhardness of the specimen was determined with a microhardness tester (DMH-2, Matsuzawa Seiki Co., Tokyo, Japan) using a Vickers diamond indenter and a 300 g load applied for 15 s. Two sites were measured at random for each specimen, and the mean value and standard deviation were obtained and compared.

#### 2.3.3. Depth of Cure

The curing depth of the specimen was determined according to ISO 4049 (2019). Each group specimen (*n* = 5) was prepared in a cylindrical Teflon mold with a length of 10 mm and a diameter of 4 mm. The top of the mold was covered with a polyester film, and the excess material was removed by pressing with a glass slide. The specimen was irradiated vertically for 20 s using a curing machine. After polymerization, the soft uncured material at the bottom of the sample was gently scraped off with a plastic spatula. The height of the polymerized material was measured using a digital vernier caliper (Mitutoyo Co., Kawasaki, Kanagawa, Japan) to an accuracy of ±0.l mm, and this value was divided by two to determine the depth of cure.

#### 2.3.4. Ion Release

Each sample was formed into a disc using a mold having a diameter and height of 10 and 2 mm, respectively. Then, samples from each group (*n* = 3) were stored in 5 mL of distilled water at 37 °C. After 24 h, water samples containing the eluted ions were collected, and the concentration of each ion was measured. Elemental analysis of the P, Ca, Na, and Zn ions released from discs was achieved by inductively coupled plasma optical emission spectrometry (ICP-OES, Optima 8300, PerkinElmer, Waltham, MA, USA).

### 2.4. Antibacterial Properties

#### 2.4.1. Inhibition Zone Tests

Bacterial analyses were carried out using *Streptococcus mutans* (*S. mutans*) (ATCC 25175) cultured in brain–heart infusion (BHI) in an incubator. One-hundred microliters of bacterial culture suspension (1 × 10^4^ cells/mL) of *S. mutans* was spread uniformly on BHI agar plates. Each material was placed in a mold with a diameter of 10 mm and a thickness of 2 mm to form disc-shaped samples. Solid samples of the test materials were placed in direct contact with the agar surface. A filter-paper disc with the same diameter as the disc-shaped sample was placed on the surface of the agar plate soaked with 20 μL of 5.25% sodium hypochlorite (NaOCl) as a positive control (PC) and 20 μL of distilled water as a negative control (NC). The plates were incubated for 24 h at 37 °C, and the inhibition zones around each sample were measured with Vernier calipers (Mitutoyo, Kawasaki, Japan) with an accuracy of 0.01 mm.

#### 2.4.2. Colony-Forming Units

A specimen was manufactured using a mold with a diameter of 10 mm and a thickness of 1 mm. A bacterial solution of 1 × 10^8^ cells/1 mL of *S. mutans* was added to the specimen, and the sample was cultured in an incubator at 37 °C. After 24 h, the specimen was placed in fresh BHI medium. Then, the samples were washed twice with medium, placed in 1 mL of BHI liquid medium, and ultrasonically cleaned (SH-2100; Saehan Ultrasonic, Seoul, Korea) for 5 min to remove bacteria physically attached to the specimen. After the bacteria had been detached, 100 mL of the separated bacterial solution was dropped on to BHI solid medium, spread, and incubated at 37 °C for 24 h. Then, the number of colony-forming units (CFU) was evaluated. Five specimens were prepared for each group, and the above process was repeated. The average values and standard deviations were calculated.

### 2.5. Statistical Analysis

All statistical analyses were performed using IBM SPSS version 23.0 (IBM Korea Inc., Seoul, Korea) for Windows with data from at least three independent experiments. Results obtained for the control and experimental groups were analyzed by one-way analysis of variance (ANOVA) followed by Tukey’s test. Statistical significance was assessed at the *p* < 0.05 level.

## 3. Results

### 3.1. Characterization of Zn-PBG

The XRD patterns of the amorphous Zn-PBG are shown in Figure 1. There are no sharp peaks, indicating the non-crystalline nature of typical glasses. Figure 2A shows the particle size distribution of the Zn-PBG powder. Zn-PBG presents a bimodal distribution, and the particle size ranges from 0.9 to 17 µm having a median diameter (*d*_50_) of 4.5 µm. The corresponding morphology of the crushed Zn-PBG particles is shown in Figure 2B. The chemical distribution of phosphorous (P), calcium (Ca) sodium (Na), zinc (Zn) and oxygen (O) was quite homogeneous (Figure 2C). Notably, there are a range of sizes and aspect ratios, and the particles are irregular and some are aggregated.

### 3.2. Flexural Strength

The flexural strength and elastic modulus of the control and Zn-PBG groups are shown in Figure 3A,B), respectively. In Figure 3A, the flexural strength of the control group is significantly higher than that of the other groups (*p* < 0.05). The 1.9, 3.8, and 5.4 wt.% Zn-PBG groups have flexural strengths of 111.3 ± 8.4, 89.6 ± 5.5, and 84.7 ± 6.8 MPa, respectively, significantly lower (*p* < 0.05) than that of the control group (130.6 ± 12.4 MPa). Nevertheless, all values are more than 80 MPa, thus meeting the ISO 4049 standard. In Figure 3B, there are no significant changes in the elastic modulus values with increase in the amount of Zn.

### 3.3. Microhardness

The means and standard deviations of the Vickers microhardness for each group are presented in Figure 3C. The results reveal that there are no significant differences in the results between the control group and the two test groups, except for the 5.4 wt.% Zn-PBG group. The hardness values of the control, 1.9, 3.8, and 5.4 wt.% Zn-PBG groups are 40.18 ± 3.72, 38.11 ± 2.79, 37.94 ± 2.29, 34.62 ± 1.24 VHN, respectively. Thus, the 5.4 wt.% Zn-PBG group has the lowest microhardness (*p* < 0.05).

### 3.4. Depth of Cure

Figure 3D shows the mean and standard deviation of the depth of cure ratio for the control and Zn-PBG groups. The mean depth of cure values of the control, 1.9, 3.8, and 5.4 wt.% Zn-PBG groups are 4.58 ± 0.10, 4.82 ± 0.03, 4.80 ± 0.09, and 4.85 ± 0.13 mm, respectively. There are no significant differences in the depth of cure between groups. Thus, the incorporation of Zn-PBG did not affect the polymerization of the composite. The depth of cure values of all groups were more than 1.5 mm and meet the ISO 4049 standard.

### 3.5. Ion Release

After 24 h, the amounts of leached P, Ca, Na, and Zn ions were determined, and the results are shown in Table 2 and Figure 4. The results indicate an increasing trend of ion release for P, Ca, Na, and Zn with increase in the amount of Zn-PBG. For all ions measured, there are significant differences between the control group and the 5.4 wt.% Zn-PBG group (*p* < 0.05). In particular, there is a significant difference in the amount of P ions released from Zn-PBG-containing samples compared to the control, even at 3.8 wt.% Zn-PBG. In addition, the amount of P ions released from the sample with 5.4 wt.% Zn-PBG was more than 19-times higher than that of the control group. The release rate of Zn ion after 24 h of the 1.9, 3.8, and 5.4 wt.% Zn-PBG groups are 0.66 × 10^−5^, 1.97 × 10^−5^, and 2.44 × 10^−5^ ppm/sec, respectively.

### 3.6. Inhibition Zone

The antibacterial activity of *S. mutans* with respect to the loading with Zn-PBG was tested using inhibition zone tests, and the results are shown in Figure 5. After 24 h of culture, the antimicrobial activity against *S. mutans* increased as the content of Zn-PBG increased, and there was no significant difference in the size of the growth inhibition zone in the negative control, control, and 1.9 wt.% Zn-PBG groups. The zone of inhibition of the 3.8 wt.% Zn-PBG group was significantly larger than that of the control, and that of the 5.4 wt.% Zn-PBG group was larger still (*p* < 0.05).

### 3.7. Colony-Forming Units

The numbers of colonies of *S. mutans* attached to the specimens were calculated, and the results are shown in Figure 6. The 1.9 wt.% Zn-PBG group yielded a similar number of CFU to the control (*p* > 0.05). For the 3.8 wt.% Zn-PBG group, the number of CFU was decreased significantly (*p* < 0.05) compared to the control. With increase in the Zn-PBG loading, bacterial inhibition increased, and the number of CFU for the 3.8 and 5.4 wt.% samples were significantly different between each other and with respect to the 1.9 wt.% sample and the control (*p* < 0.05).

## 4. Discussion

The development of a composite resin with antibacterial ability is required, and several studies have been conducted to improve the antibacterial properties of composite resins [21,22]. However, although the addition of an antimicrobial agent increases the antimicrobial activity, it can also reduce the mechanical strength [23]. In addition, the antibacterial activity is not persistent in the long term because of the release of the antibacterial substance from the composite with time [24]. Therefore, in the development of a new composite resin, the physical properties must be maintained but the antibacterial properties must be improved, especially the long-term activity [4]. In addition, bioactive materials must prevent demineralization. When a bioactive material is placed in living tissue, biochemical reactions occur, promoting bonding between the living tissues and the foreign material [20,25,26]. Thus, in this study, a flowable resin composite containing zinc phosphate glass was developed to prevent secondary caries.

First, the successful preparation of the zinc-doped phosphate glass was confirmed by XRD measurements, which provided crystal phase and structural data. The XRD patterns contained no sharp reflections, indicating that Zn-PBG is an amorphous glass [16,17]. Secondly, the bioactive flowable resin containing Zn-PBG was fabricated, and the mechanical and antimicrobial effects were evaluated.

Our first null hypothesis that “the flowable resin-composite containing Zn-PBG would not result in significant differences in the mechanical properties compared to the control” can be partially rejected. To allow clinical application, the mechanical properties of the material should be maintained [27]. There were no significant differences in the elastic modulus, microhardness, and depth of cure. This is important because the flowable resin must have mechanical properties appropriate for the mechanical stresses produced on mastication [23]. In 5.4% Zn-PBG groups, the flexural strength and microhardness were decreased compared to the control. The phosphate glass has lower mechanical properties than other glass systems because of its weak structure. Additionally, Zn-PBG particles without silane treatment might be adhered loosely to the resin matrix and act as a non-adhesive filler. Therefore, it is considered that the flexural strength and microhardness were decreased as the content of Zn-PBG increased. There were significant differences in the flexural strength results, but the mechanical strengths meet the ISO 4049 standards and match or exceed those of some commercial flowable resins [23].

The second null hypothesis that “the flowable resin-composite containing Zn-PBG would not result in significant differences in P, Ca, and Zn ion release compared to control” can be rejected. The release of P, Ca, Na, and Zn ions from the Zn-PBG-containing samples showed significant differences from the control. The Zn-PBG-containing flowable resin composite can release calcium and phosphate ions to prevent dental caries, and the silica glass filler provides the required load-bearing ability. Previous studies have investigated the remineralization of demineralized teeth using composite materials that release calcium and phosphate ions, such as composites containing hydroxyapatite, α-tricalcium phosphate, B-tricalcium phosphate, and 45S5 bioactive glass as fillers [25,27,28]. These studies revealed that the bioactive glass in the composite resin has bioactive properties and leads to the formation of tooth-like hydroxyapatite [29]. In addition, another study revealed its acid-neutralizing property [4,30]. These results indicate the possibility of the remineralization of demineralized teeth [20].

The third null hypothesis that “the flowable resin-composite containing Zn-PBG would not result in significant differences in antibacterial properties compared to control” can be rejected. Both the inhibition zone and CFU tests showed significant differences to the control and showed antibacterial activity. This antibacterial activity is thought to be related to Zn ions. Zinc is an essential mineral with a role in bacterial resistance [9,10,11], and Zn-PBG is believed to have potent antibacterial activity [31]. This effect stems from the influence of cationic ions in increasing the pH of the surroundings and also by the intracellular incursion of zinc ions, which causes disruption of the cell membrane [32]. In addition to the antibacterial activity of the zinc ions, the combined action of Ca, P, Na, and Si ions has been suggested to cause an anabolic response [33].

In this study, the first strategy to prevent secondary caries is based on the release of ions related to remineralization. Ca and P ions are released from the PBG and enhance tooth remineralization and prevent mineral loss, as previously reported [33]. The remineralization mechanism of the bioactive-glass-containing resin composite is mainly attributed to the release of ions from the bioactive glass at the composite surface, resulting in the formation of a hydroxycarbonate apatite layer on the glass surface, and the interaction of the released ions with the demineralized tissue [13]. The second strategy to prevent secondary caries is based on decreasing bacterial viability. The flowable resin containing Zn-PBG exhibited ion release resulting in antibacterial effects. Thus, this is a promising method to minimize the demineralization around restored sites. In this study, we have shown that the Zn-PBG-containing resin has appropriate chemical and physical properties for use as a dental material. It also has antibacterial properties. However, our experiments are limited by the short experimental period and the use of only one bacterial species. Therefore, a long-term follow-up study using various bacteria is necessary. In addition, further research is needed to investigate the relationship between the rate of Zn ion release and antibacterial effect over time.

## 5. Conclusions

Considering the limitations of this study, we rejected the first, second, and third null hypotheses. Further, this study has demonstrated that Zn-containing resins can have an antimicrobial effect against the adhesion of *S. mutans*. In addition, we have shown than Zn can be successfully incorporated into the resin without deteriorating the mechanical properties. These data suggest that resin can be effectively used in flowable resin composites with dental application. Zn-PBG is an antimicrobial material as a filler that can be adjusted in size. Therefore, Zn-PBG is considered to be a potential material for application as a nanomaterial.

## Figures and Tables

**Figure 1 nanomaterials-10-02311-f001:**
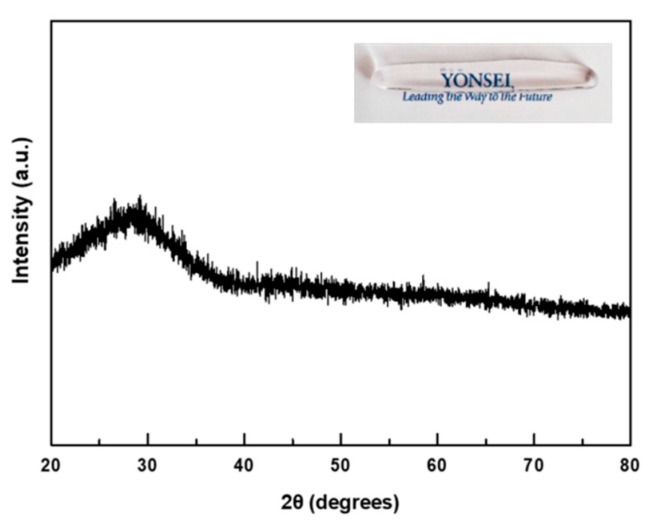
XRD patterns of the Zn-PBG powder (before mixing with flowable composite resin). The absence of crystallization peaks was confirmed on the sample, indicating the characteristic of the amorphous glass structure.

**Figure 2 nanomaterials-10-02311-f002:**
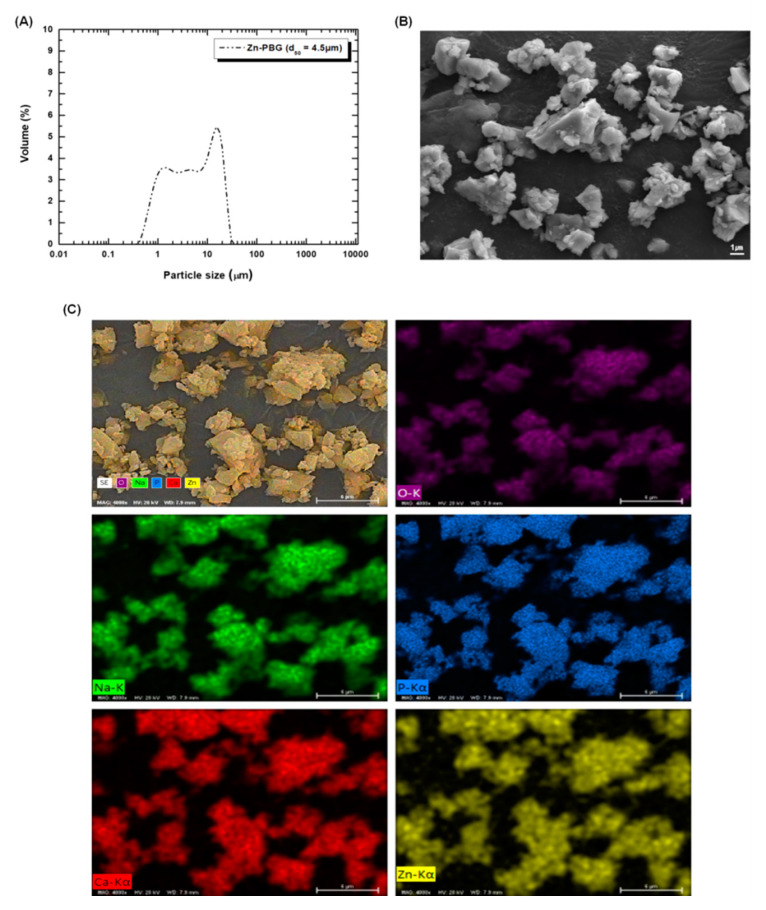
Particle size distributions (**A**), FE-SEM images of Zn-PBG powder (before mixing with flowable composite resin) (**B**) and overlap of the EDX mapping of the chemical elements distribution (**C**). Scale bar is 1 µm (**B**) and 6 µm (**C**).

**Figure 3 nanomaterials-10-02311-f003:**
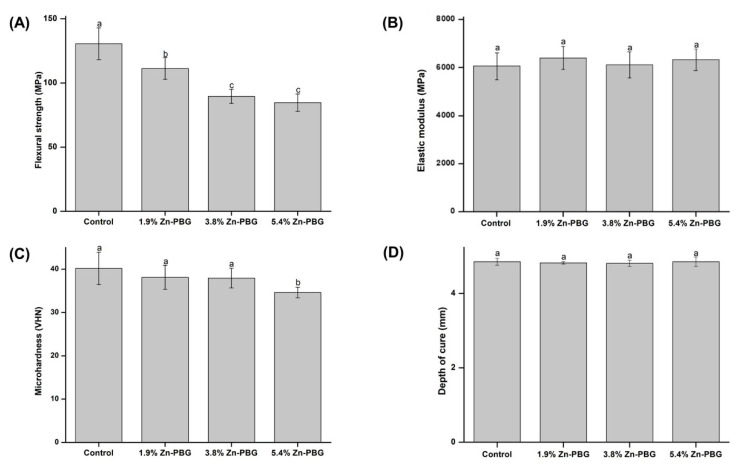
Comparison of flexural strength (**A**), elastic modulus (**B**), microhardness (**C**), and depth of cure (**D**) of samples. Letters indicate significant differences at *p* < 0.05.

**Figure 4 nanomaterials-10-02311-f004:**
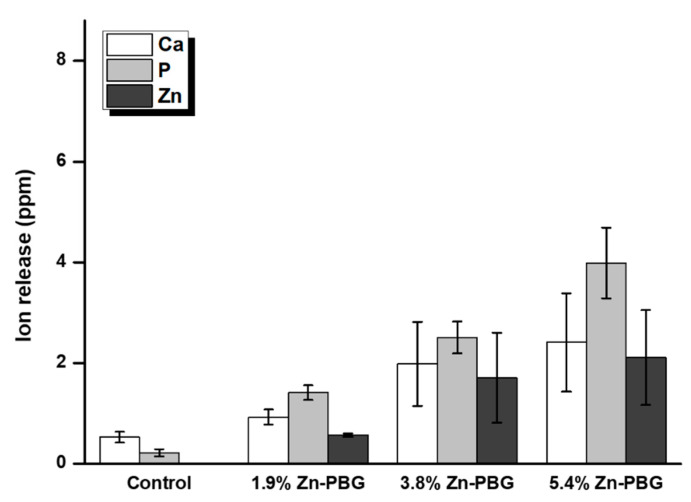
Concentrations of the major ions (Ca, P, and Zn) released from samples.

**Figure 5 nanomaterials-10-02311-f005:**
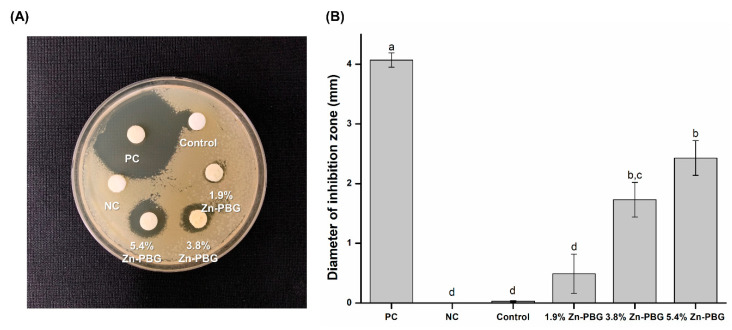
Antibacterial properties: (**A**) Photograph of inhibition zone test and (**B**) diameters of zones of inhibition on agar plates.

**Figure 6 nanomaterials-10-02311-f006:**
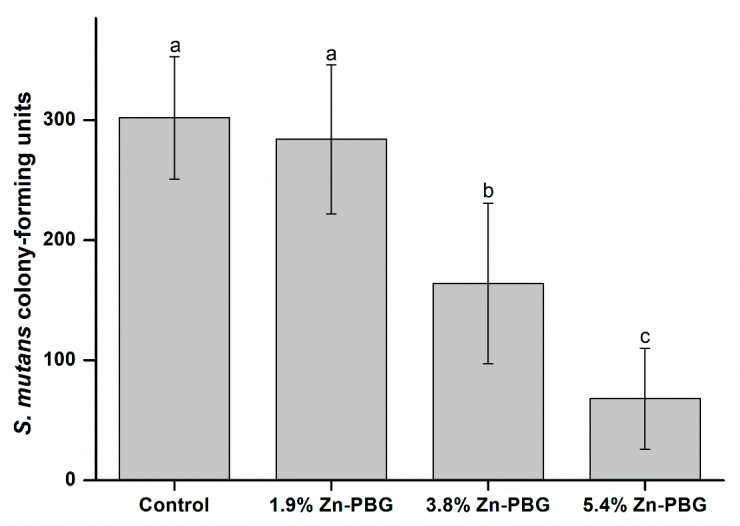
CFU counts for *S. mutans* on sample surface.

**Table 1 nanomaterials-10-02311-t001:** Control and experimental groups in the study.

Group	Group Code	Resin (wt.%)	Zn-PBG (wt.%)
1	Control	100	0.0
2	1.9 wt.% Zn-PBG	98.1	1.9
3	3.8 wt.% Zn-PBG	96.2	3.8
4	5.4 wt.% Zn-PBG	94.6	5.4

**Table 2 nanomaterials-10-02311-t002:** Released concentrations of P, Ca, Na, and Zn ions from each group (SD: standard deviation).

	Concentration (ppm) Released							
	Control		1.9 wt.% Zn-PBG		3.8 wt.% Zn-PBG		5.4 wt.% Zn-PBG	
	Mean	SD	Mean	SD	Mean	SD	Mean	SD
P	$0.22^{A}$	0.07	$1.42^{\mathrm{AB}}$	0.15	$2.51^{A}$	0.31	$3.99^{C}$	0.70
Ca	$0.53^{A}$	0.11	$0.93^{\mathrm{AB}}$	0.15	$1.98^{\mathrm{AB}}$	0.83	$2.41^{B}$	0.98
Na	$0.22^{A}$	0.06	$0.48^{\mathrm{AB}}$	0.03	$1.21^{\mathrm{AB}}$	0.61	$1.56^{B}$	0.67
Zn	−	−	$0.57^{A}$	0.04	$1.71^{B}$	0.89	$2.11^{B}$	0.94

Letters indicate significant differences at *p* < 0.05.

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
