# Peer review of "Development of a Bioactive Flowable Resin Composite Containing a Zinc-Doped Phosphate-Based Glass"

_nanomaterials, 2020, doi:10.3390/nano10112311_

Round 1
Reviewer 1 Report
In this study, the antimicrobial properties of flowable composite resins with added Zn-PBG were investigated the suitability of the flowable composite containing Zn-PBG for use in dentistry. The objective of this study was to investigate the mechanical and ion releasing properties, as well as the antibacterial activities of bioactive flowable resin composites containing Zn-PBG.
As the attached file, I have 7 questions.

Author Response
Reviewer #1 |
|
Question 1 |
Page 3, line 100 Is the content of Zn-PBG in Table 1 preparation ratios? or measured value? |
Response to Question 1 |
First of all, thank you for your detailed review and we do apologize for lack of information. We now have added more details in corresponding section as follows; Glass powder with a composition of P2O5 (42 mol.%), CaO (25.2 mol.%), Na2O (16.8 mol.%), and ZnO (16 mol.%) was prepared, and it was named Zn-PBG. Zn-PBG powder was mixed into the flowable composite resin at various weight percentages (1.9, 3.8, and 5.4 wt.%), and a flowable composite resin without Zn-PBG was used as a control. |
|
|
Question 2 |
Page 5, line 170-173 and Page 9, line 257-261 There is no detail explanation of Zn-PBG in Figure caption and sentence. What kind of Zn-PBG is indicated in Figure 1, 2(A) and 2(B)? Do the structures of all types of Zn-PBG (Group 1-4 in Table 1) have the amorphous? Is Zn-PBG really composite? Isn’t Zn-PBG the mixture of Zn, Na, and Ca? about the SEM image of Figure 2, how about the distribution of Zn atom? Did you try the observation by SEM-EDX? |
Response to Question 2 |
Thank you for your comments. According to your comments, we have revised the figure caption (Figure 1) as follows, adding detailed explanations; Figure 1. XRD patterns of the Zn-PBG. The absence of crystallization peaks was confirmed on the sample, indicating the characteristic of the amorphous glass structure. Figure 1 shows the XRD patterns of the Zn-PBG that confirmed the absence of crystallization peaks, indicating the characteristic of the amorphous glass structure. This is XRD analysis of Zn-PBG and not the composite. Composite would contain small (1.9 wt% to 5.4 wt%) proportion of Zn-PBG in resin. Also, in this experiment, glass powder with a composition of P2O5 (42 mol.%), CaO (25.2 mol.%), Na2O (16.8 mol.%), and ZnO (16 mol.%) was prepared, and it was named Zn-PBG. Samples were prepared by mixing flowable resin and Zn-PBG (1.9, 3.8, and 5.4 wt.%). These details are now added in Materials and Methods; Glass powder with a composition of P2O5 (42 mol.%), CaO (25.2 mol.%), Na2O (16.8 mol.%), and ZnO (16 mol.%) was prepared, and it was named Zn-PBG. Zn-PBG powder was mixed into the flowable composite resin at various weight percentages (1.9, 3.8, and 5.4 wt.%), and a flowable composite resin without Zn-PBG was used as a control. Additionally, we performed SEM-EDX to identify atoms including Zn and the results are as follow; Figure 2 has been now modified along with modifications of figure legends in accordance to comment above. |
|
|
Question 3 |
Page 5, line 181 Why does the addition of Zn-PBG cause a lowering of flexural strength? |
Response to Question 3 |
Thank you for your helpful comments. The lowering of flexural strength was thought to be related to the structural weakness. Also, absence of silane treatment may have the role. Hence, we have added the relevant sentence as follows in Dicussion; The phosphate glass has lower mechanical properties than other glass systems because of its weak structure. Additionally, Zn-PBG particles without silane treatment might be adhered loosely to the resin matrix and act as a non-adhesive filler. Therefore, it is considered that the flexural strength was decreased as the content of Zn-PBG increased. |
|
|
Question 4 |
Page 6, line 189-190 Why does the addition of Zn-PBG, especially 5.4 wt%, cause a lowering of microhardness? |
Response to Question 4 |
Thank you for your helpful comments. The lowering of hardness also seems to be related to the same reason as stated above for flexural strength. Hence, relevant information is now added in Discussion. |
|
|
Question 5 |
About all Figure and figure caption, The (A), (B)…. in Figure and (a), (b)…of figure caption should be matched. For example, (A) in Figure change to (a), unified to Lowercase of character. |
Response to Question 5 |
Sorry for our mistake. We now have matched upper case letters in figure and figure caption. |
|
|
Question 6 |
Page 8 Figure 5 is cut off from the manuscript. |
Response to Question 6 |
Sorry for our mistake in the production of Figure 5. Figure 5 now has been revised. |
|
|
Question 7 |
Page 8, Figure 6 How about the sustained release speed of Zn ion from composite resin? How long will the effect against antibacterial activity be maintained? Is the antibacterial activity lost immediately if the release speed of Zn ion is too fast? |
Response to Question 7 |
Thank you for your valuable comments. According to your comment on the rate of Zn release from composite resin, we’ve added details as follows; The release rate of Zn ion after 24 h of the 1.9, 3.8, and 5.4 wt.% Zn-PBG groups are 2.44 x 10-5, 1.97 x 10-5 and 0.66 x 10-5 ppm/sec, respectively. According to your recommendation, we will investigate to the long-term antibacterial activity as well as the release rate of Zn ion according period of time in further study, and we’ve added details as follows; In addition, further research is needed to investigate the relationship between the rate of Zn ion release and antibacterial effect over time. |
|
|
Reviewer 2 Report
This manuscript submitted by Myung-Jin Lee and co-workers described the development of a bioactive flowable resin composite. I think this paper is well written and the data are sound. However, a few flaws were found, and this manuscript should be revised before it can be accepted. The errors and comments are listed below:
- In figure 3B and figure 4 for P, the error bar looks too big, which means the data are not reliable. The authors should redo those experiments and provide better data.
- Lines 236-261, I feel those sentences are quite strange. The reason why/how designed those experiments should be presented in the introduction part. The discussion part is usually for data or mechanism interpretation.
- For the conclusion part, the authors had better state the “nano”material part. I feel the contents of this paper were not really relevant to the “nanomaterials” journal but close to the “coatings” journal.
Author Response
Reviewer #2 |
|
Comment 1 |
In figure 3B and figure 4 for P, the error bar looks too big, which means the data are not reliable. The authors should redo those experiments and provide better data. |
Response to Comment 1 |
Thank you for the valuable suggestions. According to your comments, we have retested elastic modulus and ion release, and Figure 3B, Figure 4 and Table 2 have been revised in the manuscript. We appreciate all your comments which we believe that they significantly improved our manuscript. |
|
|
Comment 2 |
Lines 236-261, I feel those sentences are quite strange. The reason why/how designed those experiments should be presented in the introduction part. The discussion part is usually for data or mechanism interpretation. |
Response to Comment 2 |
Thank you very much for your helpful comments. I fully agree with your opinion, and we revised to include such information in the Discussion part. |
|
|
Comment 3 |
For the conclusion part, the authors had better state the “nano”material part. I feel the contents of this paper were not really relevant to the “nanomaterials” journal but close to the “coatings” journal. |
Response to Comment 3 |
Thank you for your valuable input. According your comments, the contents are also reflected in the conclusion part. We’ve added details as follows:
Zn-PBG is an antimicrobial material as a filler that can be adjusted in size. Therefore, Zn-PBG is considered to be a potential material for application as a nanomaterial.
|
Round 2
Reviewer 1 Report
Review II of nanomaterials-99151
In this study, the antimicrobial properties of flowable composite resins with added Zn-PBG were investigated the suitability of the flowable composite containing Zn-PBG for use in dentistry. The objective of this study was to investigate the mechanical and ion releasing properties, as well as the antibacterial activities of bioactive flowable resin composites containing Zn-PBG.
The moieties in manuscript I pointed have not yet been improved. I still have 2 Questions.

Author Response
Reviewer #1 |
|
Question 1 |
Page 5, line 170-173 and Page 9, line 257-261 There is no detail explanation of Zn-PBG in Figure caption and sentence. What is Zn-PBG in XRD pattern, SEM, EDX and DLS? 1.9% Zn-PBG or 3.8% Zn-PBG or 5.4 % Zn-PBG? There is no description in Figures. |
Response to Question 1 |
Sorry for the confusion. Zn-PBG in XRD, SEM, EDX, DLS are powder before mixing with flowable resin. The properties of powders were first considered before mixing with resin at variable weight percentage. Additional detail is now included in figure legends. |
|
|
Question 2 |
You should indicate EDX Mapping instead of EDX spectrum. |
Response to Question 2 |
Thank you for your comments. EDX mapping has been now included as part of Figure 2.
|
Reviewer 2 Report
This revised manuscript has already revised according to my suggestion. I think this paper can be accepted in the present form.
Author Response
Thank you for your acceptance